# Start-Up and Performance of a Full Scale Passive System In-Cluding Biofilters for the Treatment of Fe, as and Mn in a Neutral Mine Drainage

Jérôme Jacob [ID], Catherine Joulian [ID] and Fabienne Battaglia-Brunet *

Bureau de Recherches Géologiques et Minières (BRGM), F-45060 Orléans, France; j.jacob@brgm.fr (J.J.); c.joulian@brgm.fr (C.J.)
* Correspondence: f.battaglia@brgm.fr; Tel.: +33-238643930

**Abstract:** Passive mine drainage treatment plants are the scene of many chemical and biological reactions. Here, the establishment of iron (Fe), arsenic (As), and manganese (Mn) removal was monitored immediately after the commissioning of the Lopérec (Brittany, France) passive water treatment plant, composed of aeration cascades and settling ponds followed by pozzolana biofilters. Iron and As were almost completely removed immediately after commissioning, while Mn removal took more than 28 days to reach its maximum performance. Investigations were performed during two periods presenting strong variations in feeding flow-rates: from 2.8 $m^3.h^{-1}$ to 8.6 $m^3.h^{-1}$ and from 13.2 $m^3.h^{-1}$ to 31.3 $m^3.h^{-1}$. Design flow rate was reached during the first week of the second period. Dissolved Fe and As were not affected by the decrease in residence time while Mn was only slightly affected. Microbial communities in biofilter presented similarities with those of the pond sludge, and genera including Mn-oxidizing species were detected. Proportion of bacteria carrying the *aio*A gene encoding for As(III)-oxidase enzyme increased in communities during the second period. Results suggest Mn removal is mainly associated with bio-oxidation whereas removal of Fe and As could be mainly attributed to chemical oxidation and precipitation of Fe, possibly helped by As(III) bio-oxidation.

**Keywords:** neutral mine water; passive biotreatment; full scale plant; manganese oxidation; arsenite oxidation; microbial communities; residence time

## 1. Introduction

In water draining mining works, oxidation of sulphide minerals by oxygen can generate acid mine drainage laden with high concentrations of iron (Fe), sulfate, heavy metals, and metalloids [1]. Where the ore contains sufficient carbonate rocks such as limestone or dolomite, the water may remain at a circum-neutral pH, but concentration of contaminant compounds, mainly Fe, arsenic (As), manganese (Mn), and zinc (Zn), can remain above discharge limit [2] and be detrimental to the environment, especially to aquatic plants and wildlife, but also to irrigated soils [3–5]. The toxicity of As is a well-known, but complex phenomenon. In mine water, As is mainly found in trivalent form As(III) ($HAsO_2$) and pentavalent form As(V) ($HAsO_4{}^{2-}$), the former being less mobile and less toxic [6]. The impact of an As laden mine drainage on the diversity and distribution of bacterial genes involved in As biotransformations in river waters has been demonstrated [7]. In France, environmental quality standards (EQS) derived from the Water Framework Directive (2000/60/EC) for surface water is 0.83 $\mu g.L^{-1}$. This value can be modified to take into account the local geochemical background. High concentration of Mn can be toxic for human health [8] and for the environment [9]. In France, there is currently no EQS for Mn, but predicted no-effect concentrations (PNEC) in fresh water was determined at 15 $\mu g.L^{-1}$ [10]. In the UK, the EQS for Mn is 30 $\mu g.L^{-1}$ [11]. Impacts of Mn on human health and ecosystems [12] justify the active development of efficient processes to remove this metal from mine water.

Mine drainages need to be treated to avoid environmental impacts such as loss of aquatic life or entry of heavy metals into the food chain. They have a life span that can sometimes reach hundreds of years and therefore continue to generate risks after the end of the economic activity from which they originated. It is important to develop treatment technologies as inexpensive as possible in the long run as treatments will last a long time and most often rely on public fundings. These treatments, which require little or no energy, chemical products, and maintenance, are called passive. But passive mine drainage treatment plants have a fixed size, and unlike active plants, they cannot adapt to flow rate variations, which are significantly fluctuating in mine drainages in relation to rainfall. They undergo major fluctuations in water residence time that can affect performance and lifetime of the treatment plant. For example, if the treatment process is based on fixed biomass and the flow rate is low most of the time, contaminant removal only happens near the inflow in the fixed bed, the active biomass only growing there. In this case, when the flow rate increases, the whole volume of the active fixed bed would be necessary to maintain the treatment performances, thus temporary degradation of water quality might happen. It is thus important to study passive treatment plant performances and the chemical and biological processes of metals removal under different residence times in order to better understand the limitations of the treatment plants and improve their design. In these systems, both chemical and biological phenomena may contribute to contaminant removal, but it can be difficult to determine the respective importance of each. Simultaneously observing the evolution of abundance and diversity of microbial communities and the metals and metalloids removal rates can be very helpful to better discriminate and evaluate the role of these different factors. Commissioning of a treatment plant is a privileged period to study these parameters.

In the environment, Mn is most often present in oxidation state II, III, and IV, but in anoxic mine water it is in the form of Mn(II). The main passive treatment to remove Mn in mine water is oxic rock filter [13], although wetlands can also contribute [14]. These filters can use inert media (pozzolana) or alkaline reactive media (limestone). The main steps involved in Mn removal are oxidation of Mn(II), precipitation of Mn(III/IV) minerals, and adsorption of Mn(II) [13]. These processes are frequently catalyzed by endogenous biomass present in the filter [15]. Many microorganisms, such as bacteria, fungi, and algae, can promote the oxidation of Mn. The wide diversity of Mn bio-oxidation pathways can be divided between direct oxidation by cellular components such as enzymes and indirect oxidation by microorganism metabolites and/or pH and redox changes [16]. Direct Mn oxidation to fuel the growth of chemolithoautotrophic microorganisms has been only discovered very recently [17]. It is also known that Mn precipitation is autocatalytic: Mn oxides can catalyse the oxidation and/or precipitation of Mn(II) [18]. However, the exact mechanism of Mn transformation in oxygenated conditions can follow a complex combination of steps: adsorption of Mn(II) on Mn(III) or Mn(IV), oxidation of Mn(II) to Mn(III) by oxygen or by Mn(IV), disproportionation of Mn(III) to Mn(II) and Mn(IV), and oxidation of Mn(III) to Mn(IV) [19,20]. The sorption capacity of the oxide for Mn(II) increased from 0.5 mol of Mn(II) per mole of $MnO_2$, at pH 7.5, to 2 mol of Mn(II) per mole of $MnO_2$, at pH$\sim$9 [19]. This mechanism can explain the important effect of pH on Mn removal even between pH 6.0 and 8.0 observed by Hallberg and Johnson [18] at Wheal Janes mine site.

Microbial communities play a major role in environmental As(III) oxidation processes, the abiotic reaction of As(III) with $O_2$ being very slow in a large range of pH [21–23]. Passive treatment of As-containing mine water through different technology combinations has been reported. Aerobic combined with anaerobic wetlands coupled with lime drains and aeration proved to be efficient for relatively low (<3 mg.L$^{-1}$) As concentrations [24–27]. A real scale anaerobic sulphate-reducing bacteria-based plant could treat moderately acidic (pH 5.8) effluent containing up to 50 mg.L$^{-1}$ As [28]. The oxidation and co-precipitation of Fe and As in an As-rich (100 mg.L$^{-1}$) acid mine drainage was efficient to significantly decrease As concentration at pilot scale, on site [29,30].

Bio-processes involved in As(III) and ferrous iron Fe(II) oxidation, and co-precipitation of As(V) with the formed Fe oxy-hydroxides were proposed for the development of a simple passive treatment for the neutral mine water of the Lopérec site in French Brittany [22]. More precisely, a biofilm formed with bacteria from the site clearly increased the rate of Fe(II) oxidation, As(III) oxidation, and total As removal in laboratory filter bioreactors at a residence time of 4 h. However, at this experimental step, Mn removal was not studied. In a 2-m$^3$ pilot bioreactor implemented on the site, partial Mn removal was obtained together with efficient Fe and As co-precipitation, with a residence time of 45–60 min [31]. Here, results of the monitoring of the real-size passive Lopérec treatment plant implemented site are presented. The research aimed to: (i) study the evolution of contaminants removal and biomass during the start-up of the passive treatment plant, in order to determine which major biological and geochemical processes contribute to water treatment efficiency, and (ii) compare the performances of the treatment plant at very short and very long residence times to provide feedback on the plant sizing.

## 2. Materials and Methods

### 2.1. Site

The study site is the former mining site of Lopérec in Brittany, France (48°27′2.1730″ N; −4°02′1.5645″ W), where a gold-arsenopyrite mineralization was discovered in 1987. The ore, located at the Devonian–Carboniferous transition characterized by acid and basic volcanism associated with epithermal polysulphide mineralization, consists of arsenopyrite-rich quartz veins and Fe–Ca-rich chert hosted by acid volcanic rocks (rhyolite and tuff) intercalated in calc-schist and black schist. The ore is rich in sulphide minerals such as arsenopyrite, pyrite, pyrrhotite, chalcopyrite, sphalerite, and galena [32]. Exploration work carried out between 1991 and 1992 led to the digging of a 140 m long incline. The Lopérec deposit has inferred resources of 360,000 metric tons grading 8.15 g.t$^{-1}$ of gold, but a lack of sufficient reserves led to the site being closed in 1994. The water quality of the mine drainage emerging from the incline during the present study, i.e., from April to August 2017 and from March to July 2018, is shown in Table 1.

**Table 1.** Lopérec mine drainage characteristics during the monitored period, from April to August 2017 and from March to July 2018.

| Parameters | Average | Min | Max |
|---|---|---|---|
| Flow (m$^3$.h$^{-1}$) | 16.7 | 11.6 | 33.0 |
| pH | 6.4 | 6.1 | 6.7 |
| ORP (SHE * mV) | 233 | −18 | 323 |
| Dissolved oxygen (mg.L$^{-1}$) | 0.4 | 0.0 | 1.4 |
| Temperature (°C) | 12.4 | 12.1 | 12.9 |
| Total dissolved iron (mg.L$^{-1}$) | 6.9 | 4.2 | 9.3 |
| Total dissolved manganese (µg.L$^{-1}$) | 1162 | 1027 | 1334 |
| Total dissolved arsenic (µg.L$^{-1}$) | 167 | 30 | 268 |
| As(III) (µg.L$^{-1}$) | 119 | 1 | 244 |
| As(V) (µg.L$^{-1}$) | 62 | 22 | 135 |

Note: * Standard Hydrogen Electrodes (SHE).

Mine drainage flow was measured several times at more than 50 m$^3$.h$^{-1}$ and up to 79 m$^3$.h$^{-1}$ outside this study period. The water quality objectives for Lopérec mine water are 3 mg.L$^{-1}$ for Fe, 1000 µg.L$^{-1}$ for Mn, and 100 µg.L$^{-1}$ for As (prefectural order n°2010–1324 of 13 October 2010).

### 2.2. Passive Treatment System

The Lopérec passive mine drainage treatment plant (Figure 1) consists of an aeration cascade, two settling ponds, two re-aeration cascades, and two up-flow biofilters which work in parallel. The primary aeration cascade is 10 m long, 1.5 m deep, and is composed of 10 steps. Water flow is evenly split between the two settling ponds of 300 m$^3$ (380 m$^2$) each

to give Fe(II) time to oxidize and Fe hydroxides to settle. The two re-aeration cascades are 1.5 m deep concrete manholes. They ensure that water is fully oxygenated before entering the biofilters. The two up-flow 30 m$^3$ biofilters are 1 m deep and filled with pozzolana (20–40 mm). The biofilters were inoculated with 1 m$^3$ of pozzolana from the bioreactor used during onsite pilot scale tests [31]. They were designed to remove As and Mn. It is possible to supply water to only one of the settling ponds or one of the biofilters. The system is designed for a maximum of 60 m$^3$.h$^{-1}$ of mine water flow. When the flow rate is higher, the excess flow rate by-passes the plant.

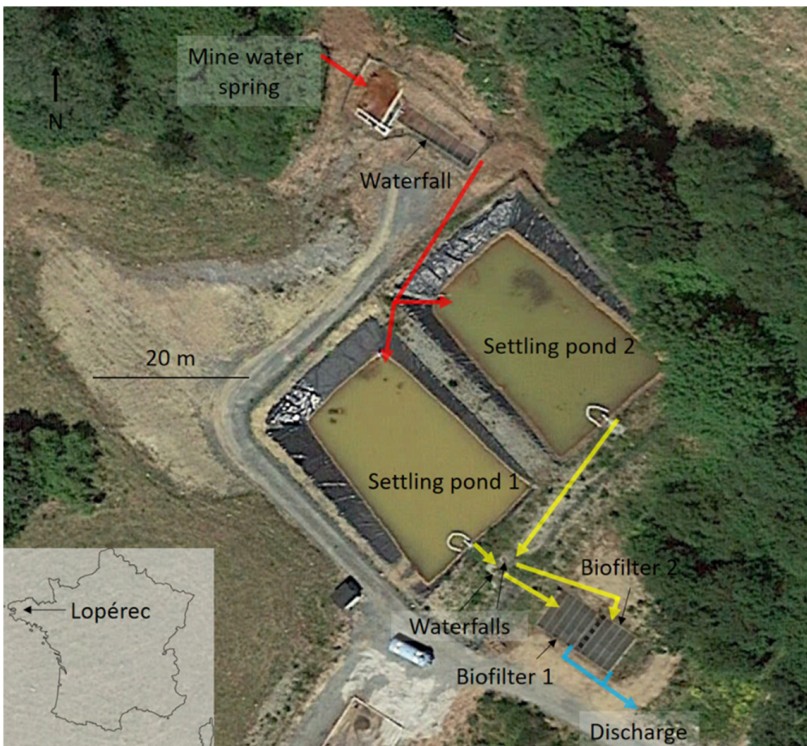

**Figure 1.** Lopérec mine drainage passive treatment system.

### 2.3. Physico-Chemical Parameters Monitoring

The Lopérec treatment system was investigated from April to August 2017 and from March to July 2018. Manual samplings and measurements were made every 2 weeks in mine drainage spring, in the outlet of settling pond 1 and in the outlet of biofilter 1 (settling pond 2 and biofilter 2 were not investigated). Additional samples were taken daily in mine drainage spring and biofilter 1 outlet with automatic samplers (ISCO 3700, Teleydyne ISCO, Lincoln, OR, USA). Settling pond 1 and biofilter 1 were fed the entire mine water flow only in 2018 to study the efficiency of Fe, As, and Mn removal at a shorter hydraulic residence time (HRT). Mine water flow was measured by an ultrasonic flow meter on a venturi flume between the water spring and the first cascade. Water flowrate at the outlet of the settling ponds were determined on a regular basis using a graduated container and a stopwatch. Actual residence time in the biofilter was verified by analysis of residence time distribution (RTD, [33]). Porosity of the fixed-bed has not been taken into account. Total suspended solids were measured by filtration of 1 L of water on weighted 0.45 μm PVCF membrane filter and drying at 40 °C for 48 h. The redox potential was measured with a silver chloride electrode WTW SenTix® ORP-T 900 and pH was measured with a WTW Sentix® 950 probe (Xylem Analytics Ltd., Letchworth, UK). Dissolved O$_2$ was measured using a FDO925 probe (Xylem Analytics Ltd., Letchworth, UK). For each manual sampling point, two water samples were taken. The first one was immediately filtered using 0.45 μm filter for analysis of dissolved Fe, As, and Mn. The second one was not filtered for analysis of total Fe, As, and Mn. Water was filtered and pre-acidified in the automatic sampler.

All samples were acidified at pH 2 with nitric acid ($HNO_3$) 67%. Fe, As, and Mn were quantified by atomic absorption spectroscopy (AAS, Varian, Palo Alto, CA, USA). As(III) and As(V) were separated using an ion exchange method [34]. Separation was performed on anionic resin (AG 1-X8, Biorad, Hercules, CA, USA). Sludges from the settling pond and the biofilter were analyzed by XRD (BRUKER D8 ADVANCE, Bruker, Germany).

### 2.4. Determination of Kinetics

Contaminants removal kinetics (k in $mg.L^{-1}.h^{-1}$ or $\mu g.L^{-1}.h^{-1}$) were calculated as follows:

$$k = \frac{C_i - C_f}{HRT}$$

where $C_i$ and $C_f$ are inlet and outlet contaminant concentration ($mg.L^{-1}$ or $\mu g.L^{-1}$) and *HRT* is hydraulic residence time (h). Porosity of the fixed-bed was not taken into account.

### 2.5. Biomolecular Monitoring

Samples were taken in sterile PolyEthylene (PE) bottles. Pozzolana pieces were taken in the upper layer of biofilter 1 (10 cm deep). They were kept at 5 °C until they were crushed in a sterile mortar for DNA extraction. Sludge from the settling pond 1 was taken near the outlet of unit and kept at 5 °C. Microbial DNA was extracted using the FastDNA™ Spin Kit for Soil (MP Biomedicals, Irvine, CA, USA), using a FastPrep-24™ instrument at a speed of $5 m.s^{-1}$ for 30 s. DNA extract concentration was measured using the Quantifluor dsDNA sample kit and the Quantus fluorimeter (Promega, Madison, WI, USA).

Real-time quantitative PCR (qPCR) was used to determine abundances of genes as proxy of total bacteria (16S rRNA gene) and As(III)-oxidizing bacteria (*aio*A, encoding arsenite oxidase catalytic subunit). The reaction mixture contained: SSO Advanced Supermix (BioRad, Hercules, CA, USA), supplemented with 100 ng of T4GP32, primers pairs 341F/534R (0.4 µM, 16S rRNA gene) or aoxBM4-1F/aoxBM2-1R (0.3 µM, *aio*A gene), and 1 to 6.7 ng of extracted DNA. qPCR programs are described in Table S1. Standard curves were obtained from serial 10-fold dilutions of a linear plasmid carrying the targeted gene. No-template controls were run for each qPCR assay. All measures were performed in duplicate, in a CFX Connect Real-Time PCR Detection system (Bio-Rad), and data were analysed with the CFX Manager 3.1 software (Bio-Rad). Results are expressed as ratio and gene copies/g of sludge or pozzolana.

Microbial diversity was assessed by 16S rRNA gene metabarcoding. Amplicon libraries and sequences were generated by INRAE Transfert (Narbonne, France). Briefly, the V4-V5 region of the gene coding 16S rRNA (Bacteria and Archaea) was amplified using the barcoded, universal primer set 515WF/918WR [35]. PCR reactions were performed using AccuStart II PCR ToughMix kit and cleaned (HighPrep PCR beads, Mokascience). Pools were submitted for sequencing on Illumina MiSeq instrument at GeT-PlaGe (Auzeville, France). Sequences were processed using the FROGS (v.3.2) bioinformatics pipeline [36], implemented into the Genotoul plateform of the Galaxy server of Toulouse. Fastq paired reads were merged with Vsearch software, and clustering into OTU (Operational Taxonomic Unit) with Swarm and an aggregation distance clustering of 1. Chimera and OTU with a proportion less than 0.0005% of all sequences were removed. Taxonomic affiliation was performed using BLAST and the Silva 138.1 database for 16S rRNA gene sequences. The FROGS pipeline implemented Phyloseq R package was used for OTU structure visualisation, rarefaction curves, and alpha diversity index calculations. Chao 1 index (non-parametric measurements) estimates the number of unobserved species from those observed 1 or 2 times while the Shannon and invSimpson diversity indexes give respectively a measure of heterogeneity of microbial community and the probability that two randomly picked individuals do not belong to the same species. The raw datasets are available on the European Nucleotide Archive system under project accession number PRJEB53214.

## 3. Results and Discussion

### 3.1. Evolution of HRT in the Treatment Plant

HRT in the treatment plant depends on natural variation of the spring water flow rate. In the settling pond and in the biofilter, between the beginning of spring and summer, water flow rate decreased from 8.6 $m^3.h^{-1}$ to 2.8 $m^3.h^{-1}$ in 2017 and from 31.3 $m^3.h^{-1}$ to 13.2 $m^3.h^{-1}$ in 2018. This led to a gradual increase in HRT in both years (Figure 2). Residence time was lower in spring and summer 2018, because the entire mine drainage flow was directed into a single settling pond and biofilter to study the influence of shorter HRT (compared to 2017) without the effect of known variation of water chemistry in winter when the flowrate is naturally higher. The first two flowrates measured in 2018 are close to the design flow of the system (maximum flowrate before by-pass). Residence times reached 10 h in the settling pond and 1 h in the biofilter.

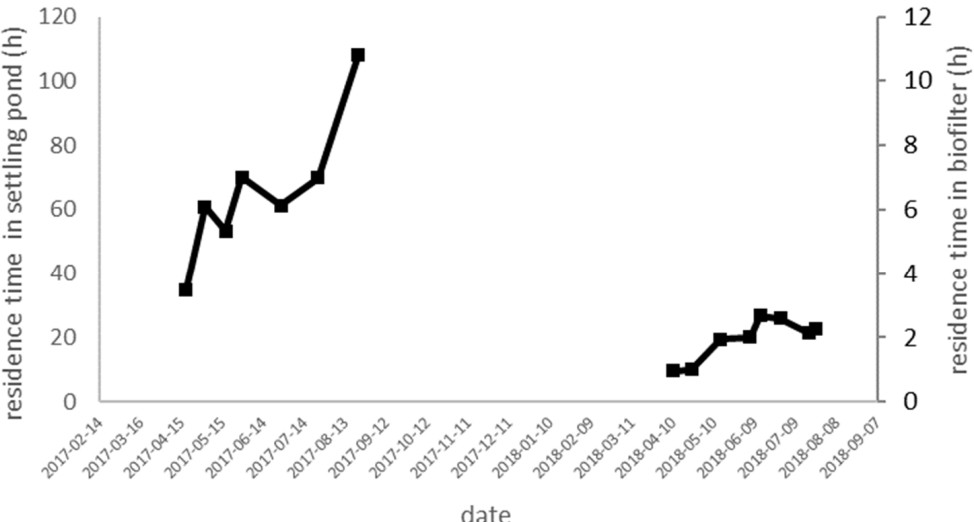

**Figure 2.** Evolution of hydraulic residence time (HRT) in the settling pond and in the biofilter in 2017 and in 2018. The values for the settling pond and the biofilter are directly proportional (factor of 10) because they both depend on the same flow-rate and on fixed volumes. The left *Y*-axis gives the scale for the settling pond and the right *Y*-axis gives the scale for the biofilter.

### 3.2. As and Fe Removal

During both 2017 and 2018, a seasonal increase in Fe and As concentrations in the mine drainage was observed (Figure 3). This evolution, observed every year at the Lopérec site, is attributed to a decrease in dilution of the mine drainage by groundwater in connection with the decrease in rainfall during spring. Dissolved Fe and As were always entirely removed in the settling pond, even when reducing the HRT in 2018 by directing the flow in a single settling pond and biofilter. The first sample, taken about 48 h after commissioning, is the only one showing dissolved Fe removal below 97% in the settling pond. Removal of dissolved Fe and As in the settling pond was always high (>95%) despite the shorter HRT in 2018, showing a high efficiency of their removal in the treatment plant. Arsenic removal was always above 93% in the settling pond. Lopérec treatment system average performance in % are presented in Table S2. These results show that HRT in the settling pond is much longer than needed to remove dissolved Fe and As and that the removal processes of Fe and As take place very quickly upon commissioning.

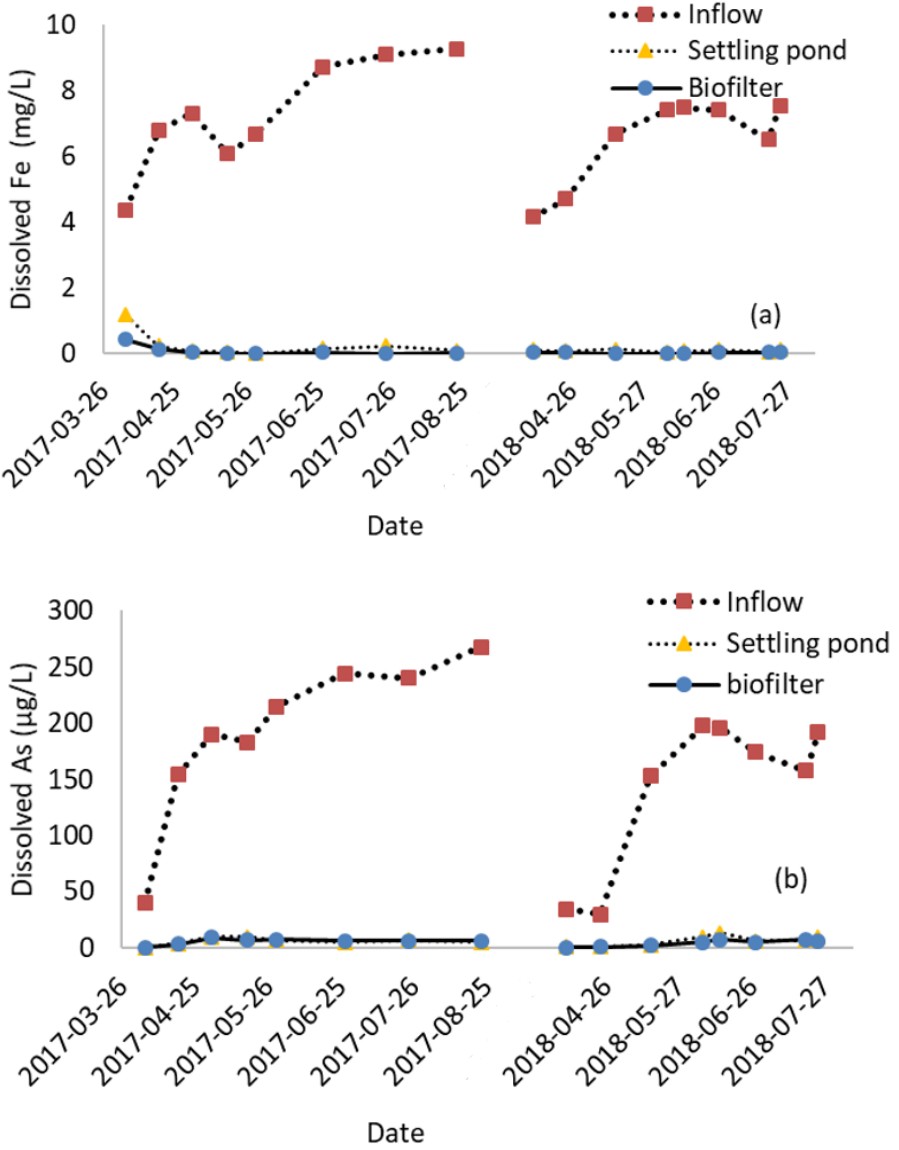

**Figure 3.** Dissolved Fe (**a**) and As (**b**) concentrations.

Iron was certainly removed by oxidation of dissolved Fe(II) in Fe(III) which precipitated and settled. The cascade plays an important role in the oxidation of Fe(II) by oxygenating the water and stripping dissolved carbon dioxide which increase pH according to Equation (1) [37]:

$$H^+ + HCO_3^- \rightarrow CO_2 + H_2O \tag{1}$$

According to the review of Morgan and Lahav [38] the oxidation rate of ferrous iron can be described by Equation (2) (in units of $mol.L^{-1}.min^{-1}$):

$$-\frac{d\left[Fe^{2+}\right]}{dt} = 6 \times 10^{-5}\left[Fe^{2+}\right] + 1.7\left[Fe(OH)^+\right] + 4.3 \times 10^5\left[Fe(OH)_2^0\right] \tag{2}$$

At pH values below 4, the $Fe^{2+}$ concentration dominates, and the rate is independent of pH. At pH > 5, $Fe(OH)_2^0$ determines the rate because it is far more readily oxidized than both $Fe^{2+}$ and $Fe(OH)^+$. Between pH 5 and 8 $Fe(OH)_2^0$, the concentration rises steeply with pH and the overall oxidation rate increases accordingly. At pH values > 8, concentration of $Fe(OH)_2^0$ no longer varies with pH and the oxidation rate is again independent of pH. According to this simple kinetic model (no biological activity, no interaction with other

elements), it should be noted that between pH 5 and 8, the oxidation rate doubles for every 0.15 pH increase.

Dissolved oxygen, redox potential, and pH all increased between the mine drainage spring and the outlet of the cascade (Table 2).

**Table 2.** Water chemistry in the Lopérec mine drainage treatment system.

| Year | Sampling Point | pH | ORP (SHE mV) | Dissolved Oxygen (mg.L$^{-1}$) |
|---|---|---|---|---|
| 2017 | Inflow | 6.2 | 194 | 0.3 |
| | Cascade | 6.7 | 353 | 7.8 |
| | Settling pond | 7.0 | 366 | 8.6 |
| | Biofilter | 7.0 | 298 | 8.4 |
| 2018 | Inflow | 6.6 | 287 | 0.4 |
| | Cascade | 7.0 | 310 | 7.9 |
| | Settling pond | 7.3 | 372 | 9.8 |
| | Biofilter | 7.3 | 388 | 10.2 |

Given the circumneutral pH, Fe(II) oxidation is probably chemical, but it is also possible that chemical or biological catalysis reactions take place on the surface of the cascade steps. This hypothesis might be supported by the relatively high Fe concentration in the first sample, suggesting that the Fe removal process is not immediately established. Arsenic removal in the settling pond is certainly due to either co-precipitation or adsorption to the surface of Fe minerals [39]. Given the near-neutral pH of the water, the low sulfate concentration (53 mg.L$^{-1}$), and the rapid rate of Fe oxidation and precipitation, ferrihydrite or goethite are probably the main phases formed in the settling pond [40,41]. The precipitates produced in laboratory bioreactors treating synthetic Lopérec mine water were composed of 2-line ferrihydrites with a few diffraction peaks indicating the presence of crystallized Fe(III) products, mainly hematite and goethite [22]. Ferrihydrite and goethite are both know to remove As by co-precipitation and adsorption in acidic waters [42] and by adsorption in neutral waters [43]. In contrast with Fe, As was immediately removed, certainly because the Fe/As ratio is more than enough to remove all As even if Fe precipitation was not complete. Average dissolved As/Fe ratio in Lopérec water is 22.9 mg.g$^{-1}$. Rait et al. [43] have shown that by adsorption only, Fe can remove arsenic up to 12 mg.g$^{-1}$ in circumneutral water and up to 74 mg.g$^{-1}$ in acidic water. Sekula et al. [44] showed that ochreous precipitate from a circumneutral mine drainage contained up to 69 mg.g$^{-1}$ As.

During the first monitoring period (2017), an increase in dissolved As(V) proportion was observed at the outlet of the settling pond and biofilter (Figure 4a), which shows that As oxidation took more than 14 days to be established. Results for the pond and biofilter correspond to the few μg.L$^{-1}$ of As not removed (less than 3% of influent dissolved As). However, they suggest that the maximum efficiency of the plant for As(III) oxidation was reached at the beginning of May 2017, i.e., one month after the plant started to treat the mine water. This delay is a few higher than that observed in the on-site column tests: 12 days for total As(III) oxidation [22]. Arsenic speciation was only punctually measured in 2018 (three dates, not shown); however, they confirmed the persistence of As(V) as major species in the pond and biofilter outlets. Biological oxidation of As(III) by microorganisms of the Lopérec site was already demonstrated [22], and confirmed here by the presence of *aio*A genes, specific to As(III)-oxidizing bacteria, quantified in the settling pond sludge and in the biofilter (Figure 4b). Their proportion within the bacterial community (measured by the *aio*A/16S rRNA genes ration) increased about one month after commissioning, and this delay could be the time needed to develop sufficient As(III)-oxidizing bacterial biomass in order to oxidize more than 97% of As(III). As(III) is more mobile than As(V) because of its neutral form ($H_3AsO_3$) being the dominant form at neutral pH [45]. Nevertheless, these results show that at the Lopérec site with an HRT long enough and a high Fe/As ratio in

the spring water, the target As removal efficiency can be reached independently from total As(III) oxidation.

**Figure 4.** Dissolved As(V) proportion (**a**), and *aio*A/16S rRNA genes ratio (**b**) during the 2017 monitoring period.

The settling pond was designed to remove Fe while the biofilter was designed to remove As and Mn based on the laboratory scale and pilot scale tests [22,30]. In the actual treatment plant, Fe and As are both efficiently removed in the settling pond stage, probably because the HRT (at least 10 h) is much longer than during the laboratory (1 h) on-site pilot scale (3 h) experiments. The oxygenation of the water by the cascade certainly also plays an important role, which could not be observed in the experiments at laboratory and pilot scales as they were not equipped with a proper water aeration system.

### 3.3. Mn Removal

During both 2017 and 2018, there was a seasonal decrease of dissolved Mn concentrations in the mine drainage (Figure 5). Manganese was removed to a small extent (7–20%) in the settling pond, but more extensively (34–97%) in the biofilter. Manganese removal in the settling pond was constant and occurred quickly (except in the first sample). The Mn removal rates in the settling pond are consistent with the PhreeqC calculations of Mn co-precipitation by Fe made by Rose et al. [46]. In the biofilter, Mn removal was very low for the first 28 days, then Mn concentration at the biofilter outlet decreased rapidly until day 43. Finally, Mn concentration continued to decrease slowly. The observed delay in reaching maximum Mn removal rate can be explained by the colonization of the fixed-bed by a Mn-oxidising biomass, by the accumulation of the first layer of autocatalytic Mn oxide, or both processes. The duration of this start-up phase is quite comparable with that observed in a non-inoculated quartzite fixed-bed (33 days) at laboratory scale [47], whereas a parallel system with inoculated dolomite and $MnO_2$ fixed-bed removed 50% of influent Mn from the first day. Similarly, the fixed-bed described in Jacob et al. [48] and composed of limestone partially inoculated with sediments from the Lopérec biofilter immediately entirely removed Mn, which suggests the importance of limestone, $MnO_2$, and bacterial biomass on the duration of the start-up phase. Present results and literature data indicate that both bio-oxidation and autocatalytic phenomena are involved in Mn removal. Biological oxidation of Mn is certain, as there is initially no Mn oxide and as at this pH (7.0) chemical oxidation alone is extremely slow [49,50]. Auto-catalytic oxidation of Mn is very likely given the rapid decrease in Mn concentration from day 28 to day 43. The difference between these two phenomena cannot be precisely distinguished, but given the shape of the curve, the following scenario may be proposed. During the first 28 days, the biomass started to colonize the biofilter and to produce Mn oxide. From this moment, the autocatalytic reaction could have started as Mn concentration decreased very quickly with the accumulation of Mn oxides in the filter. However, something seems to limit Mn

removal since it slowly decreased from day 43 onwards, while a significant concentration of dissolved Mn remained in the water.

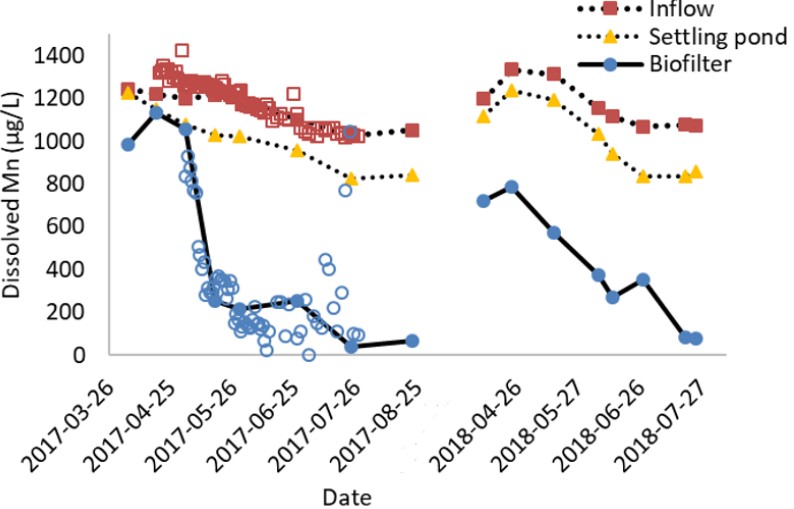

**Figure 5.** Dissolved manganese concentration.

Manganese removal ratio (in %) decreased with decreasing HRT, while Mn removal kinetic (in $\mu g.L^{-1}.h^{-1}$) increased (Figure 6a). When the HRT was at the minimum of 1 h (design flow), the Mn removal rate reached a maximum of 403 $\mu g.L^{-1}.h^{-1}$, but the Mn removal ratio was only 30%. Those results are different from those of Luan et al. [51] who reported a linear relationship between HRT and Mn(II) removal rate. This could be explained by the Lopérec biofilters design as they were filled with pozzolana and not limestone, and thus pH is independent of HRT. Results show that Mn removal in the Lopérec biofilters is a first order reaction as summarized by Morgan and Stumm [19] for abiotic Equation (3):

$$\frac{d[Mn^{2+}]}{dt} = k_0\left[Mn^{2+}\right] + k_1\left[Mn^{2+}\right]\left[MnO_2\right] \tag{3}$$

where $k_0 = 4 \times 10^{12}$ M$^{-3}$[O$_2$·Aq][OH$^-$]$^2$ and $k_1 = 10^{18}$[O$_2$·Aq][OH$^-$]$^2$.

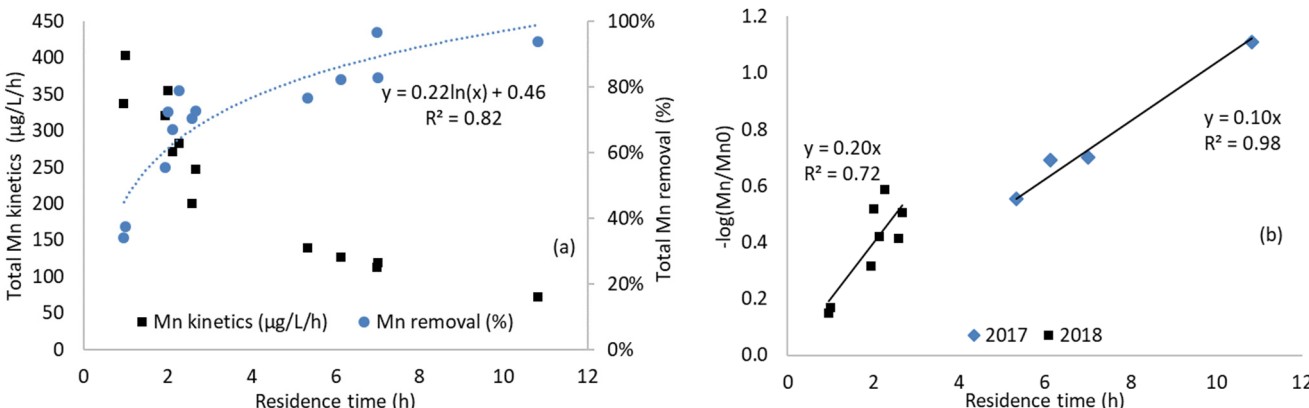

**Figure 6.** Mn removal kinetics (based on total reactor volume) and total Mn removal (not filtered) vs. HRT in biofilter (**a**). The first three values at the start of the biofilter functioning have been removed; and linear regression of -log(Mn/Mn0) versus HRT (intercept through 0 imposed), y = first order Mn(II) oxidation rate constant (**b**).

If pH is circumneutral, then the term $k_0[Mn^{2+}]$ is negligible and if Mn oxidizing bacteria are active, then according to [46] the relation can be written as Equation (4):

$$\frac{d[Mn^{2+}]}{dt} = k_1\left[Mn^{2+}\right][MnO_2] + k_2\left[Mn^{2+}\right][bact] \tag{4}$$

To oxidize 1 mg.L$^{-1}$ of Mn(II) to Mn(III) or Mn(IV), between 0.15 and 0.30 mg.L$^{-1}$ of dissolved $O_2$ are needed [52]. Given dissolved $O_2$ concentrations measured at the inlet and outlet of the biofilter and Mn concentrations, it can be safely assumed that dissolved $O_2$ is not a limiting factor of the Mn removal kinetics. Thus, the limitation could be linked to bacterial biomass and activity, or to the amount of Mn oxide available for auto-catalyze. The concentrations of Mn oxide and bacteria are certainly not homogeneously distributed in the pozzolana bed. The lower layer of the up-flow biofilter is certainly enriched in bacteria and $MnO_2$ compared to the upper part, because it is the part where the Mn(II) concentration is the highest. Moreover, most of the time, water flow rate is much lower than the design flow: thus, 80% of the time the mine drainage flow rate is less than 27 m$^3$.h$^{-1}$, i.e., a HRT of more than 2h15 min (Figure S1). When the flow rate increases the overall Mn(II)dissolved/MnO$_2$ solid ratio decreases, and Aguiar et al. [53] has demonstrated the importance of this ratio for Mn removal.

On the other hand, Mn oxide concentration is not constant over time, obviously during the start-up but also in the long term. In 2017, it was calculated that biofilter 1 accumulated around 182 kg of $MnO_2$. The surface of Mn oxide available to catalyse precipitation does not increase as fast as the mass, especially when there is an increase in the thickness of the $MnO_2$ layer on the surface of the pozzolana. Nevertheless, an increasing performance of the biofilter can be expected over time, and it can be seen that in Lopérec the first-order rate constant for Mn removal calculated for 2018 (0.2 h$^{-1}$) is twice those of 2017 (0.1 h$^{-1}$) (Figure 6b).

Despite low Mn concentrations and the use of pozzolana instead of limestone, the average and maximum Mn removal kinetics measured at Lopérec, 4.1 g.m$^{-2}$.d$^{-1}$ and 9.7 g.m$^{-2}$.d$^{-1}$, respectively, are within the range of values reported by Rose et al. [46] (1.2 to 13.2 g.m$^{-2}$.d$^{-1}$) and Luan et al. [13] ($-1.0$ to 14 g.m$^{-2}$.d$^{-1}$). This is certainly due to the vertical flow, which allows a deep bed and the pH of water, relatively high. Johnson and Younger [47] reported a much higher removal rate (60 g.m$^{-2}$.d$^{-1}$), probably because of the very high pH of their water (between 8.0 and 8.2).

*3.4. Suspended Solids*

The Fe and As precipitates are mainly retained in the settling pond and Mn precipitates are accumulated in the biofilter. However, a small part of Fe and As precipitates are transported with the treated water flow in the form of suspended solid (Table 3). After the settling pond, Fe and As are almost exclusively in particulate form and Fe accounts for the majority of suspended solids. When HRT was greater than 20 h, the average As removal of the plant was 80%, of which 65% was retained in the settling pond (Figures 7 and S2). When HRT was 10 h, the average As removal decreased to 57%, of which 29% was retained in the settling pond. The results are similar for Fe.

**Table 3.** Suspended solid concentrations and contribution (in %) of the solid phases of Fe, As, and Mn to the fluxes of elements transported in water.

| Parameters | TSS (mg.L$^{-1}$) | Fe$_{>0.45\mu m}$ (mg.L$^{-1}$) | Fe$_{>0.45\mu m}$ (%) | As$_{>0.45\mu m}$ (%) | Mn$_{>0.45\mu m}$ (%) | Fe$_P$ */TSS ** (%) |
|---|---|---|---|---|---|---|
| Inflow | n/a | 2.2 | 14 | 26 | 0 | 59 |
| Settling pond | 4.4 | 2.6 | 94 | 92 | 2 | 69 |
| Biofilter | 2.1 | 1.4 | 97 | 87 | 6 | |

Notes: * Fep: particulate Fe; ** TSS: total suspended solids.

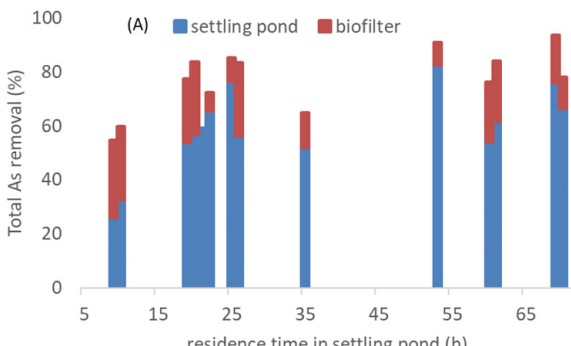
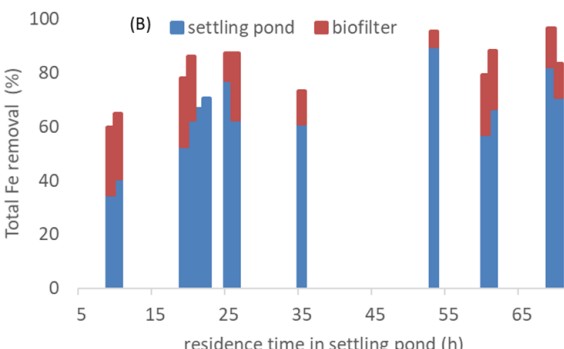

**Figure 7.** Total As (**A**) and Fe (**B**) removal in the settling pond and in the biofilter.

The efficiency of settling decreased sharply as the HRT approaches design flow. Settling is the limiting step compared to oxidation and precipitation of Fe. While HRT in the settling pond is much higher than necessary for the removal of dissolved Fe and As, it is suited for the removal of particulate Fe and As at design flow because precipitates removal decreases sharply as water flow rate approaches the design flow rate. XRD analysis of the sludge from the biofilter (Figure S3) showed the presence of chrysotile, quartz, muscovite, and albite, but did not indicate any well crystallized Fe, As, or Mn phases. However, one or more unidentified amorphous phases were also present. Sludge composition is detailed in Table S3.

*3.5. Bacterial Communities*

The composition of the bacterial communities colonizing the biofilter and the settling pond sludge evolved temporally (Figure 8). In April and May 2018, the bacterial communities of the biofilter were closely related to the bacterial communities of the settling pond sludge at the same date (Figure 8a). The bacterial communities colonizing the biofilter in 2017 and 2018, and the settling pond sludge in 2018, were dominated by *Cyanobacteria* (Figure 8b). In the bacterial communities of full-scale biofilter, three genera counting Mn-oxidizing strains represented more than 2% of the sequences, including three previously found in the laboratory biofilter [22]: *Pseudomonas* [54], *Flavobacterium* [55], and *Nitrospira* [17]. *Hyphomicrobium*, another bacterial genus associated with Mn(II) biooxidation [55,56], was present in all samples and represented 0.5 to 1.2% of the sequences. Well-known Fe(II) oxidizers were only present in very low proportion: members of the *Gallionellaceae* family represented exceptionally 1.5% of the total sequences in a biofilter sample (June 2018), and less than 1% in all the other samples. The well-known Fe(II) and Mn(II) oxidizing genus *Leptothrix* was also present in all samples but represented only 0.2 to 0.7% of the sequences. Genera of which some species are known to oxidize As were also present: at all dates for *Flavobacterium* [57], as one main OTU in June 2017 for the cyanobacterial genus *Anabaena* [58], or more sporadically for *Pseudomonas* [59].

In the laboratory pozzolana biofilter implemented at the early steps of the bioprocess development, the biofilm inoculated with the bacteria of the Lopérec site water was composed of a diverse population of microorganisms, affiliated to eight phylogenetic clades within the Gram-negative phylum of *Proteobacteria* and to one clade within the Bacteroidetes [22]. The phylotypes that were identified by 16S rRNA genes cloning/sequencing technique, were closely related to the species *Pseudomonas rhodesiae*, *Curvibacter gracilis*, *Ralstonia* sp., *Sphingomonas* capsulate, and *Methylobacterium fujisawaense*. Furthermore, two phylotypes were more distantly affiliated to *Gallionella ferruginea* and *Methylophilus methylotrophus* and a phylotype formed a sister branch to the *Herbaspirillum-Janthinobacterium* group. Finally, the only phylotype not belonging to the *Proteobacteria* but to the *Bacteroidetes* phylum was somewhat related to *Flavobacterium* spp.

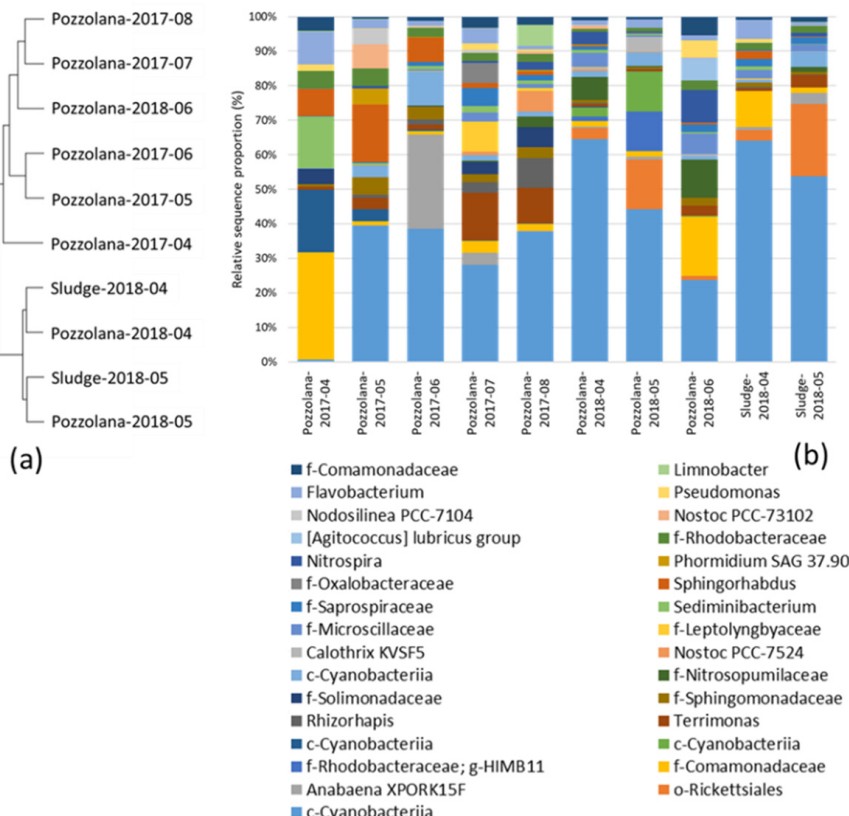

**Figure 8.** Composition of the bacterial communities in the Lopérec treatment plant. (**a**) Hierarchical clustering of the samples based on Bray–Curtis dissimilarity matrix (Ward diagram); (**b**) OTUs representing more than 2% of the sequences in at least one sample. When not identified at the genus level, higher taxonomic ranks are indicated by letters f (family), o (order), or class (c).

Here, 16S rRNA gene metabarcoding gave access to a more detailed bacterial diversity. One main difference between early laboratory experiments and full-scale treatment plant is the presence of *Cyanobacteria*, which is linked to the major role of light (during the days) as energy source, for the bacterial communities of the settling pond, that were transported with water, then retrieved in the biofilter. This phototrophic biomass could feed heterotrophic microorganisms, including some Mn(II)-oxidizers, by providing organic matter. *Cyanobacteria* might also contribute, but marginally, to $O_2$ production, because efficient aeration was obtained in the cascades. They could also help indirectly Mn(II) oxidation through consuming dissolved $CO_2$ thus contributing to increase the water pH.

## 4. Conclusions

The full-scale passive mine water treatment plant of the Lopérec site proved to be efficient for oxidation and removal of dissolved Fe, As, and Mn for a large range of HRT. The primary cascade and settling pond rapidly allowed complete oxidation and elimination of dissolved Fe and As after the plant commissioning, whereas the downstream biofilter showed an increased efficiency for dissolved Mn removal during the 2017 and 2018 monitoring periods. Mechanisms involved in the treatment could include chemical oxidation of Fe(II), co-precipitation of Fe(III), As(III), and As(V), bacterial oxidation of As(III) and Mn(II), and autocatalysis of Mn(II) oxidation by Mn(IV). Biological analyses support the hypothesis of bio-oxidation of As(III) and Mn(II), and potential direct or indirect contribution of a large proportion of the bacterial communities in the settling pond and in the biofilter, to the overall treatment efficiency.

**Supplementary Materials:** The following supporting information can be downloaded at: https://www.mdpi.com/article/10.3390/w14121963/s1, Table S1. Target genes, primers and qPCR programs. Figure S1: Cumulated frequency of Lopérec mine water flowrate; Table S2: Lopérec treatment system performance (dissolved) title; Table S3: Sludge composition. Lopérec. Figure S2: Total Fe and As in settling pond and biofilter. Figure S3: XRD spectrum of the sludge from the biofilter.

**Author Contributions:** J.J., conceptualization, project administration, methodology, writing—original draft preparation; C.J., investigation; writing—original draft preparation, review and editing; F.B.-B., writing—original draft preparation; review and editing; All authors have read and agreed to the published version of the manuscript.

**Funding:** Financial support for these studies was provided by BRGM SA within the framework of the relinquishment of its mining rights and the BRGM PDEV VITAMINE.

**Institutional Review Board Statement:** Not applicable.

**Informed Consent Statement:** Not applicable.

**Data Availability Statement:** Data are available upon request to the corresponding author.

**Acknowledgments:** The authors would like to thank the past and present project team members: Florian Koch, Dominique Breeze, Mickael Charron, Gael Bellenfant, Jean-François Brunet, Elise Decouchon, and Bernard Lamouille.

**Conflicts of Interest:** The authors declare no conflict of interest.

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
