# Peer review of "Start-Up and Performance of a Full Scale Passive System In-Cluding Biofilters for the Treatment of Fe, as and Mn in a Neutral Mine Drainage"

_water, doi:10.3390/w14121963_

Round 1

Reviewer 1 Report

The presented work is interesting, relevant and well done. To improve the text of the manuscript for readers, it is necessary to make some corrections and additions.

The following comments are below.

  • The names of the genes should be written in italics. For example, on line 18 and see in the text.
  • Key words. It is better to be consistent and write “residence time” with capital R.
  • In the "Materials and methods" section, all devices and kits must be indicated by the company that issued them, the city and the country.
  • Line 164. Please, provide a reference on the Residence Time Distribution (RTD).
  • Line 209 and Table S1. What primers have been used in the study?
  • Table 2. What does the abbreviation “SHE” mean? The article will be read by researchers with different experiences, not all of them can immediately understand the details. Therefore, it is necessary to explain in more detail.
  • S3. It is recommended to make the legend for this drawing (top right) larger. A very small image is difficult to read.

Reviewer 2 Report

The paper presents some interesting findings on the passive treatment of an actual mining site contaminated with  Fe, As, and Mn, supported by physicochemical and molecular microbial community data analyses. However, the major weakness of the paper is its overall data presentation and the quality of writing. Please refer to the attached file.

Round 2

Reviewer 2 Report

None